# Current Evidence on the Ocular Surface Microbiota and Related Diseases

**DOI:** 10.3390/microorganisms8071033

**Published:** 2020-07-13

**Authors:** Francesco Petrillo, Danilo Pignataro, Maria Annunziata Lavano, Biagio Santella, Veronica Folliero, Carla Zannella, Carlo Astarita, Caterina Gagliano, Gianluigi Franci, Teresio Avitabile, Marilena Galdiero

**Affiliations:** 1Section of Ophthalmology, University Hospital “Policlinico-Vittorio Emanuele”, 95123 Catania, Italy; francescopetrillo09@gmail.com (F.P.); t.avitabile@unict.it (T.A.); 2Section of Microbiology and Virology, University Hospital “Luigi Vanvitelli”, 80138 Naples, Italy; danilopignataro89@gmail.com (D.P.); a.lavano1989@gmail.com (M.A.L.); bi.santella@gmail.com (B.S.); 3Department of Experimental Medicine, University of Campania “Luigi Vanvitelli”, 80138 Naples, Italy; veronica.folliero@unicampania.it (V.F.); carla.zannella@unicampania.it (C.Z.); 4Sbarro Institute for Cancer Research and Molecular Medicine, Department of Biology, College of Science and Technology, Temple University, Philadelphia, PA 19122, USA; carloastarita@gmail.com; 5Section of Ocular Immunology and Rare Diseases, University Hospital “Policlinico-Vittorio Emanuele”, 95123 Catania, Italy; caterina_gagliano@hotmail.com; 6Department of Medicine, Surgery and Dentistry “Scuola Medica Salernitana”, University of Salerno, 84081 Baronissi (S.A.), Italy; gfranci@unisa.it

**Keywords:** microbiota, ocular surface, ophthalmic diseases

## Abstract

The ocular surface microbiota refers to the resident non-pathogenic microorganisms that colonize conjunctiva and cornea. Several studies have shown that ocular surface epithelial cells can respond selectively to specific components of ocular pathogenic bacteria by producing pro-inflammatory cytokines and, in contrast, they do not respond to non-pathogenic bacteria, thus supporting the colonization by a real microbiota. However, the analysis of the ocular microbiome composition is essential for understanding the pathophysiology of various ophthalmic diseases. In this scenario, the first studies, which used microbiological culture techniques, reported a less diverse profile of the ocular microbiota compared with that recently discovered using new molecular-based methods. Indeed, until a few years ago, the microbiota of the ocular surface appeared to be dominated by Gram-positive and a few Gram-negative bacteria, as well as some fungal strains. In contrast, genomics has nowadays detected a remarkable diversity in the ocular surface microorganisms. Furthermore, recent studies suggest that the microbiota of other areas of the body, such as the gut and oral microbiota, are involved in the pathophysiology of several ophthalmic diseases. The aim of the present study is to highlight the current evidence on the ocular surface microbiota to better understand it and to investigate its potential role in the development of ophthalmic diseases.

## 1. Introduction

The ocular surface is constantly exposed to the external environment and therefore it is vulnerable to contamination with microbes. However, throughout evolution, various microbes, especially bacteria, have colonized the ocular surface as commensals, forming its microbiota. Although the ocular surface is in constant contact with these commensal bacteria, the epithelial cells of cornea and conjunctiva do not give rise to an inflammatory response in healthy subjects, suggesting the existence of an innate immune response of the ocular surface epithelium that allows the colonization by a commensal microbiota [1]. The ocular microbiota has a role in maintaining the homeostasis of the ocular surface, preventing the colonization by pathogenic species. The alteration of this subtle ecosystem has been associated with a variety of pro-inflammatory states. The analysis of the ocular microbiome composition is crucial for understanding the pathophysiology of various ophthalmic diseases. Traditional culture techniques have several limitations in the detection and/or identification of uncharacterized bacteria of exogenous origin [2,3,4]. On the contrary, molecular biological techniques, such as next-generation sequencing of 16S rDNA, compensate for diagnostic culture techniques in the diagnosis of infectious diseases [5,6]. The purpose of this review is to summarize the current evidence on the ocular surface microbiota and to investigate how alterations of the ocular microbiota are associated with the development of ophthalmic disease.

## 2. Eye Structure and Innate Immunity 

Structurally, the eye consists of three layers enclosing various anatomical structures. From the outside to the inside, they are: (i) the fibrous tunic comprising cornea and sclera; (ii) the vascular tunic or uvea, that consists of the choroid, ciliary body, and iris, and (iii) the retina, which receives its blood supply from the vessels of the choroid as well as the retinal vessels. Between these three layers, there are the anterior and the posterior chambers, the lens, and the vitreous cavity [7]. The ocular surface, comprised of cornea, sclera, and its overlying tissue. the conjunctiva, is exposed to microorganisms and protects the inner parts of the eye from external microbial invasion. Resident conjunctiva lymphocytes, plasma cells, neutrophils, and dendritic cells can generate an inflammatory response and produce antibodies against potential pathogenic environmental microorganisms. Another important component of the eye is the nasolacrimal drainage system, which serves as a conduit for the eye fluids, called tears. Most humans blink about 12 times per minute. The blink reflex distributes tears as a film, known as “tear film”, over the ocular surface and is very important as it protects the eye against foreign objects that are contaminated with microbes. In fact, tears have many antimicrobial components such as lactoferrin, defensins, and lysozyme, that help to prevent colonization by pathogens. In addition, mucins facilitate the removal of microbes from the eye surface [8,9,10,11]. The ocular surface consists of cornea and conjunctiva that are exposed to the external environment and therefore they are very vulnerable to contamination with microbes (Figure 1). 

However, throughout evolution, various microbes, especially bacteria, have colonized the ocular surface as commensals, forming the ocular microbiota. Several studies have shown that the epithelial cells of cornea and conjunctiva can respond selectively to specific components of ocular pathogenic bacteria by producing pro-inflammatory cytokines and, in contrast, they do not respond to non-pathogenic bacteria, suggesting the existence and the involvement of an innate immune response of the ocular surface epithelium that allows the colonization of a commensal microbiota [13]. The innate immune system recognizes the commensal and the potential pathogen preserved antigens, termed “Microbe-Associated Molecular Patterns” (MAMPs) and “Pathogen-Associated Molecular Patterns” (PAMPs) respectively, through particular receptors called “Pattern Recognition Receptors” (PRRs). The link between PAMPs and PRRs results in cellular activation and production of proinflammatory cytokines, chemokines, and interferons, which act as an alarm signal for the immune system in the presence of an attack. PAMPs are considered real “molecular signatures” of the various pathologic agents. Among the different PRRs, the most important are Toll-Like Receptors (TLRs) and NOD-like receptors (NLRs) [14,15]. In human medicine, there are at least 13 different TLRs, and each has its own specificity [16]. The TLR family comprises transmembrane receptors expressed on different cells of the immune system and on epithelial, endothelial, and parenchymatous cells, which are able to bind different microbial ligands. In particular, TLR-4 receptor recognizes lipopolysaccharide (LPS), a component of the external membrane of Gram-negative bacteria, while TRL-2 intercepts the teichoic and lipoteichoic acids of Gram-positive bacteria and, also, the peptidoglycan of both Gram-positive and Gram-negative bacteria; flagellin, a bacterial flagellar protein, is recognized by TRL-5; double-stranded and single-stranded viral RNA bind to TRL-3 and TRL-7/TRL-8, respectively, and, finally, TRL-9 recognizes unmethylated CpG DNA motifs. The recognition of PAMPs by TLRs leads to the expression of different pro-inflammatory cytokines, chemokines, growth factors, and adhesion molecules, which activate the effector functions of the innate immune system cells (phagocytosis, oxygen radical production). As a result, a rapid inflammatory response characterized by the recruitment of leukocytes and antigen-presenting cells (APCs), such as dendritic cells or macrophages, occurs at the site of infection to eliminate the invaders [17]. The TLRs localized on APCs constitute an essential link between innate and specific immunity: the stimulation of these receptors on APC leads to the activation and priming of still naive T lymphocytes, triggering the specific immune response. The activation of TLRs at the ocular level, however, can lead to inflammation that can compromise visual integrity. Indeed, the activation is appropriate if it is directed against disabling pathogens, but it can become inappropriate if directed to the normal microbial flora. A recent study has shown that the human corneal epithelium expresses only TLR-2 and TLR-4 at the intracellular level and not on the surface. The last observation suggests that this variation, compared to the normal TRLs expression on the surface, allows a condition of “immune silence” that prevents unnecessary inflammatory responses to the normal bacterial flora [18]. Other hypotheses have been formulated to explain the lack of response of the immune system to commensal microbes. Some authors suggested that membrane TLR might be inactive or may not be expressed at a protein level; others indicated the existence of several molecules, such as TOLLIP, MyD88 short, SIGIRR, that can block TLR signaling at different levels [19].

## 3. Characterization and Role of the Normal Ocular Microbiota

The stable presence of commensal bacteria at the ocular surface level has long been a subject of debate due to the constant washing by the tear film and to the antimicrobial nature of ocular secretions. However, clinical studies have indicated that fungal infections are correlated with the topical use of ophthalmic antibiotics, suggesting that the alteration of the positive interaction between the immune ocular system and microorganisms of the normal eye’s bacterial flora exposes the eye to colonization by pathogens [20]. Using culture-based methods, it was observed that the ocular surface, as any other mucosal and cutaneous surface, shows an abundant microbial flora, consisting of Gram-positive and Gram-negative microorganisms. The genera *Staphylococcus*, *Corynebacterium*, *Streptococcus*, *Propionibacterium*, and *Micrococcus* are commensal Gram-positive bacteria present in low numbers in the eyelids, conjunctiva, and tear film [21]. These bacteria originate from the skin and colonize the ocular surface immediately after birth. This ecosystem remains relatively stable throughout life, unless it is altered by antibiotic treatment, surgical interventions, infections, or other problems (e.g., use of contact lenses in human medicine). Gram-negative bacteria, such as *Haemophilus*, *Pseudomonas*, and *Neisseria* genera, and fungi are less common, but they may be present in healthy individuals [22]. However, traditional culture methods allow the observation of only a fraction of the ocular microbiota due to the inability to detect cultivatable and slow-growing bacteria through this type of methodologies [3,4,5]. The latest generation of techniques in molecular biology, such as next-generation sequencing of bacterial 16S rRNA, has exceeded the limits imposed by traditional culture-based techniques, allowing a more extensive and realistic characterization of the diversity of the ocular surface microbiota. In 2009, researchers of the Bascom Palmer Eye Institute initiated the Ocular Microbiome Project. This project aimed to investigate diversity and community structure, identifying novel and constitutively present species and analyzing the functional role of the virtually uncharacterized ocular surface microbiome. This project had high relevance to human health because disease-causing pathogens were still unidentified for the majority of ocular surface infections and acute inflammation cases. As a result, new and previously uncharacterized pathogens were discovered on the ocular surface. In 2011, using the sequencing of the 16S rRNA gene, Dong et al. classified the bacteria of four subjects’ total conjunctival swab DNA in 5 phyla and 59 distinct genera 3. Among the 59 genera, 12 were consistently present in the conjunctive of all examined subjects, suggesting that these genera could represent a “putative core” of conjunctival microbiota and that other genera could be transient, depending on other factors such as environment, lifestyle, and physiologic differences [23]. Genomics, compared to culture-based analysis, detected a remarkable diversity in the composition of conjunctival commensal microorganisms, with an average of 221 species of bacteria per subject. In this study, it was found that the most abundant phyla were represented by Proteobacteria (64%), Actinobacteria (19.6%), and Firmicutes (3.9%). The presumptive core of the conjunctival microbiota was formed of the following genera: *Pseudomonas*, *Propionibacterium*, *Bradyrhizobium*, *Corynebacterium*, *Acinetobacter*, *Brevundimonas*, *Staphylococci*, *Aquabacterium*, *Sphingomonas*, *Streptococcus*, *Streptophyta*, and *Methylobacterium* [3]. The first five of these were the most abundant, accounting for 58% of DNA sequence reads. In 2016, at the Shandong Eye Institute, Huang at al. analyzed 31 DNA conjunctival samples from healthy adult humans using Illumina high-throughput sequencing to determine diverse taxa. At the genus level, 16S rRNA gene sequencing reading identified 25 bacterial phyla and 526 distinct genera, with an average of 158.8 ± 41.04. Ten of the 25 phyla—*Proteobacteria, Firmicutes, Actinobacteria, Bacteroidetes, Fusobacteria, Deinococcus-Thermus, Cyanobacteria/Chloroplast, Candidatus Saccharibacteria, Acidobacteria*, and *Spirochaetes*—were the most abundant. Among the 526 genera, 24 were ubiquitous in all examined subjects, and 10 of these—*Corynebacteria*, *Pseudomonas*, *Staphylococcus*, *Acinetobacter*, *Streptococcus*, *Millisia*, *Anaerococcus*, *Finegoldia*, *Simonsiella*, and *Veillonella*—were identified as common ocular bacteria in most subjects, forming the core of the ocular microbiota [24]. In another study about the conjunctival microbiome in healthy eyes of 105 participants residents in Gambia, Zhou et al. classified the bacteria in 610 genera belonging to 22 phyla through deep sequencing of V1–V3 hypervariable regions of the bacterial 16S rRNA gene. In this report, the predominant phyla were represented by *Actinobacteria, Proteobacteria,* and *Firmicutes*. At the genus level, among the 610 genera, *Corynebacterium*, *Streptococcus*, *Propionibacterium*, *Bacillus*, *Staphylococcus*, and *Ralsontia* were common to all participants. The prevalence of *Actinobacteria, Proteobacteria, and Firmicutes* phyla was shown in the above study [25]. At the genus level, both Dong et al., Zhou et al., and Zhou et al. described an abundance of *Propionibacteria*. However, according to Zhou et al., *Pseudomonas* was the least common, with less than 1% relative abundance, whereas according to Dong et al. and Huang et al., it was a component of the core of conjunctival microbiota (Figure 2).

These differences are due to the different methods of collection of the samples, the different compositions of the conjunctival microbiota of the analyzed subjects, and the trimming and deionization processes. Consistent with the microbiome data of other areas of the body, such as the epidermis, the ocular microbiome composition appears to have a vertical stratification at the genus level [26]. Indeed, opportunistic and environmental microorganisms, which represent transient species of the ocular surface, such as *Rothia*, *Herbaspirillum*, *Leptothrichia*, and *Rhizobium*, can be isolated by tamponing the ocular surface with a light pressure. On the contrary, *Staphylococci*, *Cornyebacteriae*, and *Proteobacteria* can be isolated by performing a “deeper” sampling. Therefore, a complete swabbing at different levels is necessary to obtain a precise characterization of the ocular surface microbiome diversity [3]. The role of the ocular microbiota is very important and is similar to that of the other organism’s districts (e.g., skin, intestine, and upper respiratory tract) [27]. The commensal bacterial flora communicates with the epithelial and immune cells and coordinates different functions aimed at the maintenance of homeostasis and local well-being, such as preservation of barrier function, inhibition of apoptosis and inflammation, acceleration of wound healing, competitive exclusion of potential pathogens and maintenance of immunotolerance.

## 4. Effects of Age, Sex, Ethnicity, and Geographic Location on the Ocular Microbiota Composition

Studies conducted on the ocular microbiota have demonstrated changes from birth to adulthood. At birth, the conjunctiva microbiota is very similar to that of the uterine cervix, composed mainly of *Streptococci*, coagulase-negative *Staphylococci*, and *Propionibacterium*. Two days after birth, the ocular microbiota further changes, resulting composed of *Staphilococcus epidermidis*, *Escherichia coli*, and *Staphilococcus aureus*, and continues to evolve up to pediatric age, when the microorganisms isolated from the conjunctiva are similar to those observed in adults: *Streptococcus* spp., *Micrococcus* spp., *S. aureus*, and *Corynebacterium* spp. are the most representative [28,29,30,31]. Zhou and Cavuoto described that the conjunctival microbiota of subjects under the age of 10 showed a richness (absolute number of taxa present) and a Shannon diversity index (relative number and abundance of each taxon) significantly higher compared to that of older subjects [32]. In contrast, Wen et al., analyzed the composition of the ocular microbiota in subjects within an age group between 28 and 84 years, founding a greater diversity of Shannon index values in the older subjects than in the young ones. Such contradictions could be justified by differences in hygienic behaviors, state of immunity, and interpersonal contacts [33]. Sex has less influence than age on the ocular microbiota composition. In fact, several studies have shown that there is no difference between males and females, according to Wen et al., which did not find differences at the level of phyla but only at the genus level, with a decrease of *P. acnes* and *S. epidermidis* from males to females and an increase of *E.coli* in females [33,34]. The composition of the ocular microbiota seems also to be influenced by seasonal changes, with a reduction in the richness and diversity of microorganisms in the wet season compared to the dry one. On the contrary, geographic position and ethnicity do not influence the composition of the ocular microbiota [35].

## 5. Ocular Microbiota Changes in Ophthalmic Diseases

The ocular surface microbiota plays an important role in the maintenance of local homeostasis and in the prevention of the pathogenic species proliferation. Many authors have suggested that alterations of the normal ocular microbial flora are related to several disease states, such as blepharitis, conjunctivitis, keratitis, trachoma, and dry eye syndrome (DED) [36,37,38,39]. Regarding blepharitis, Lee et al. observed a change in the microbial composition, with an increase in the amount of *S. aureus*, *Streptophyta* spp., *Corynebacterium* spp., and *Enterobacter* spp. and a decrease in *Propionibacterium* spp. compared to healthy controls [36]. In addition, *Chlamydia trachomatis* infections have been associated with reduced bacterial diversity and with an increase in the *Corynebacterium* and *Streptococcus* genera [25]. Wearing contact lenses is a possible risk factor for the development of microbial keratitis and other inflammatory eye conditions. In a study conducted by Zhang and al., the microbiota of the ocular surface of lens wearers was compared with that of non-lens wearers using 16S rRNA gene sequencing. In non-lens wearers, there was a greater wealth of *Haemophilus* spp., *Neisseria* spp., *Lactobacillus* spp., *Streptococcus* spp., coagulase-negative *staphylococci*, and *Rothia* and *Corynebacterium* spp. and a lower abundance of *Pseudomonas* spp., *Acinetobacter* spp., and *Methylobacterium* spp. compared to the lens-wearers’ flora, which was richer in *Methylobacterium* spp., *Acinetobacter* spp., and *Pseudomonas* spp. and less abundant in *Haemophilus* spp., coagulase-negative *staphylococci*, *Streptococcus* spp., and *Corynebacterium* spp. These results suggest that wearing contact lenses alters the eye microbiota, making it more similar to the skin microbiota [40,41]. *Pseudomonas aeruginosa* is one of the Gram-negative pathogens most frequently isolated from bacterial keratitis, a severe condition that can rapidly lead to the formation of descemetocele and even corneal perforation and endophthalmitis. This condition requires an adequate therapeutic approach that also takes into consideration the resistance possessed by *P. aeruginosa* towards common antimicrobial agents [41]. Recently, some authors have demonstrated the contribution of the ocular surface microbiome in regulating the possibility to induce an infectious keratitis by *P. aeruginosa*. The results of these studies have shown that the presence of a healthy ocular microbiome strengthens the innate ocular immune barrier, significantly increasing the concentrations of immune effectors in the tear film, including IgA and complement proteins [42]. The authors also conducted in vivo experiments on Swiss Webster (SW) mice, usually resistant to *P. aeruginosa*-induced keratitis, which instead became sensitive after their ocular microbiome had been altered. The protective immunity, in fact, was subsequently re-established by colonizing the ocular surface of the mice with coagulase-negative *Staphylococci*, previously isolated from conjunctival tampons. Therefore, these data underlined the role of the microbiome in the regulation of ocular sensitivity to keratitis and it are particularly important in light of the increasing isolation of multi-resistant *P. aeruginosa* (multi-drug resistant, MDR) strains in patients suffering from ocular infections [38].

In this scenario, the phenotypic characterization of *P. aeruginosa* MDR, isolated from a patient with Stevens–Johnson syndrome, is a recent development. The strain called VRFPA04 showed a highly resistant and virulent nature and caused a significant loss of visual acuity in the patient, despite an appropriate antibiotic therapy. VRFPA04 was resistant to beta-lactam antibiotics (penicillin, cephalosporins, carbapenems), aminoglycosides, and quinolones, and sensitive only to aztreonam and fourth-generation cephalosporins. A study published in *Immunity* by US and Japanese researchers analyzed the ability of a specific bacterium called *Corynebacterium mastitis*, to firmly colonize the conjunctiva by increasing resistance to pathogens. In the presence of ocular infections, the researchers showed that this bacterium, a human skin commensal microorganism found in rats, can stimulate the production of interleukin-17 by conjunctival T cells, allowing the recruitment of a greater number of neutrophils. Conversely, germ-free or gentamicin-treated mice showed a reduction in T lymphocytes and in S100A8, an ocular surface anti-microbial peptide that is normally found in the murine tear film, becoming more susceptible to *P. aeruginosa* and *Candida albicans* infections. The ability to permanently colonize the eye is not a property of all bacteria, even if they belong to the same genus. Indeed, the same repeated experiment with *Corynebacterium bovis* and *Corynebacterium glutamicus* gave negative outcomes. This research is only a first step, and the relationship between the eye and bacterial commensals must be deepened in the future [43]. Many studies have shown how different microorganisms can also contribute to the development of ocular neoplasms. Examples are the human papillomavirus, cause of human conjunctival papilloma, which has been associated with squamous cell carcinoma; the HIV virus, associated with conjunctival squamous cell carcinoma; the herpes virus, which plays a role in the development of Kaposi’s conjunctival sarcoma; and, also, *Helicobacter pylori* and the hepatitis C virus, associated with ocular adnexal mucosa-associated lymphoid tissue lymphomas [21]. In summary, the alteration of the normal commensal flora of the ocular surface, due to antibiotics or other external factors, can induce colonization of the ocular surface by opportunistic pathogens, increasing the risk of ocular neoplasms [44].

## 6. Association of Other Body Sites’ Microbiome and Ophthalmic Diseases

Recent studies suggest that the microbiome of other body sites may have a role in the development of some ophthalmic diseases. Two interesting studies have connected the oral and intestinal microbiota with the development or deterioration of different ophthalmic diseases. Regarding the oral microbiota, it was observed that subjects affected by glaucoma presented a bacterial count much higher than that of control subjects. The increase of the number of bacteria residing in the oral cavity could be related to the activation of microglia in the retina and in the optic nerve, with the consequent neurodegeneration of the optic nerve [45]. Other ophthalmic diseases were also related to immuno-mediated conditions. In particular, some of these diseases, such as uveitis, have been associated with alterations of the normal intestinal flora [46]. Indeed, as observed in animal models of experimental autoimmune uveitis, the administration of oral antibiotics that disturb the intestinal microbiota led to a significant reduction of uveitis [47]. This could be explained with a reduction of the inflammatory state associated with the use of antibiotics. The alteration of the intestinal commensal flora increases the number of regulatory T lymphocytes in lymphoid tissues as well as in the eyes, leading to the attenuation of uveitis [48]. In confirmation of what has just been said, in addition to uveitis, other eye diseases, such as scleritis, episcleritis, and keratitis, have often been observed in association with inflammatory bowel diseases (IBD), such as Crohn’s disease, caused by polygenic and environmental factors, including intestinal dysbiosis. The alteration of the intestinal microbiota also plays an important role in the development of various pathologies of the posterior eye [49]. Recent studies, conducted in mice, have highlighted how a low-sugar diet led to the development of metabolites capable of protecting the eye from the development of characteristics related to age-related macular degeneration (AMD). On the other hand, a diet rich in sugars causes an alteration of the intestinal microbiota in mice, with the appearance of characteristics similar to those of dry AMD [50]. Differences in the microbiota of the ocular surface were also found in subjects with type 2 diabetes mellitus compared to healthy controls. In fact, subjects with diabetes had a greater abundance of *Acinetobacter* [51] (Table 1).

This difference could be related to the effect of diabetes on the immune function. Alterations of the gut microbiota in the presence of type 1 and type 2 diabetes and following the use of metformin could contribute to the progression of diabetic retinopathy [52,53,54]. It is necessary to improve the studies on the microbiome of the ocular surface to better understand the microbiome’s role in the genesis of ocular pathologies and, above all, to allow the development of new therapeutic strategies, also based on probiotics, for the treatment and prevention of ophthalmic diseases.

## 7. Antibiotic Therapy Alters the Ocular Surface Microbiota

Ophthalmic antibiotics are used to treat and prevent a variety of infectious and inflammatory conditions. Many studies have shown that antibiotics can negatively alter the microbiota of the eye’s surface, potentially contributing to the opportunistic invasion of pathogenic species and eye diseases [55]. Sarita et al. evaluated the effects of a prolonged exposure to topical antibiotics, such as azithromycin, gatifloxacin, moxifloxacin, ofloxacin, on the ocular microbiota of 24 patients. This study reported a substantial change of the eye flora following antibiotic treatment. In particular, in patients without treatment, *S. epidermidis* and *S. aureus* represented 54.5% and 18.2% of the flora, respectively. After treatment with azithromycin, *S. epidermidis* and *S. aureus* constituted 90.9% and 4.5% of the flora, respectively. In patients treated with fluoroquinolones, 63.4% and 13% of *S. epidermidis* and *S. aureus* were detected, compared to controls in which they represented 45.7% and 6.5% of the flora, respectively. Furthermore, fluoroquinolone treatment reduced the percentage of Gram-negative species from 8.7% to 1.6% [56]. Ozkan et al. evaluated the effect of a topical antibiotic treatment with tobramycin on the eye microbiota during the use of contact lenses. The treatment determined the reduction of the commensal flora, in particular of Gram-positive bacteria. Similar results were obtained after treatment with moxifloxacin. This study showed a reduction of Gram-positive bacterial species and no alteration of Gram-negative bacteria [57]. Many studies have described the effects of the repeated exposure to topical antibiotics in microbial-resistance models [58]. Yin et al. investigated the effect of a repeated intravitreal administration of moxifloxacin on the conjunctival microbiota. The repeated use of this antibiotic was linked to increased antibiotic resistance of the microbiota. In particular, they observed that the average minimum inhibitory concentration (MIC) levels increased by 20% in the treated group with respect to the control group [59]. The use of antibiotics causes changes in the ocular microbiota and promotes the development of drug resistance. The ocular microbiome influences ocular homeostasis, and its alteration induced by antibiotics influences the commensal species that colonize the eye’s surface, causing an imbalance in favor of pathogenic species and increasing the risk of eye infection [21]. The studies mentioned above collectively suggest that the use of antibiotics has a measurable and immediate influence, bringing out resistant bacterial strains. Eye infections caused by MDR strains are more difficult to treat and can cause increased morbidity. As evidence of this, an experimental study showed that endophthalmitis caused by MDR *S. epidermidis* produced greater inflammation and faster destruction of the eye tissue than those caused by antibiotic-sensitive strains [60]. In response to the increase in antibiotic resistance in ophthalmology, which is documented worldwide through surveillance programs, important therapeutic alternatives are coming, including lysine, probiotics, bacteriophages, and antimicrobial peptides [61,62,63].

## 8. New Therapeutic Strategies in Ophthalmic Diseases

In the last decade, several studies have emerged on the introduction of probiotics and prebiotics in the therapy of several pathologies. Probiotics are referred to as “live microbes that, when administered in sufficient quantities, give a benefit to host health” (FAO Working Group/WHO 2002). Bacterial species used as probiotics mainly belong to the genera *Lactobacillus* and *Bifidobacterium* [64,65,66]. Prebiotics, on the other hand, are non-digestible fermentable oligosaccharides, such as fructooligosaccharide (FOS) and galactooligosaccharide, which promote the growth of probiotic bacteria, inducing the development of a healthy microbiota [67,68,69]. Researchers are evaluating the commercial feasibility of pre–probiotics developed specifically for eye care. Currently, the therapy for bacterial and allergic conjunctivitis involves the use of eye drops containing antibiotics and antihistamines or cortisone. The use of anti-inflammatory and antihistamine drugs causes unwanted side effects, such as high intraocular pressure, corneal complications, and changes in the ocular microbiota, with increased antibiotic-resistant pathogenic strains [70]. As an alternative, a probiotic therapy would promote a rapid and stable restoration of the healthy eye microbiota, faster anti-inflammatory and antihistamine effects, and an antagonistic action towards pathogens, avoiding the use of antibiotics and the resulted antibiotic resistance. The application of microbiome modulation technologies also in ophthalmology and eye care has recently been published in some studies. Based on the idea of the eye as an ecosystem in which inflammation and autoimmunity are linked to the microbiota, Chisari et al. performed a pilot study to assess the effects of probiotic supplementation (*Bifidobacterium lactis* and *Bifidobacterium bifidum*) on the lacrimal film in patients with DED [71]. After 30 days of treatment, there were statistically significant improvements in the Schirmer test (measurement of tear secretion rate) and tear break time (a measure of the stability of the tear film), as well as a reduction in the colonization by *S. aureus* at the eye area level.

As probiotics, for topical use, several substances such as quercetin (a flavonoid) were evaluated. Oh et al. examined the use of 0.5% locally applied quercetin eye drops on the eye surface of mice with DED, reporting a significant increase in tear volume after 10 days and in the restoration of smooth corneal surfaces and higher density of caliph cells [72]. Similar to quercetin, Vela et al. examined the effects of resveratrol, alone and in combination with quercetin, in mouse models, demonstrating a reduction in the clinical signs of DED, with better corneal coloration, and anti-inflammatory effects, suggesting their topical application for the treatment of DED [73]. In 2016, Chisari et al. conducted a clinical study to assess the effect of the consumption of a combination of probiotics (*Saccharomyces boulardii* MUCL 53837 and *Enterococcus faecium* LMG S-28935) on the tear film of people suffering from DED. Their results suggested that this combination of probiotics may reduce some signs and symptoms of DED, while modulating bowel function [74]. Another study demonstrated the effect of a combination of pre-probiotics on dry eyes, reporting a significant improvement in clinical symptoms after 4 and 8 weeks of treatment [75]. Kim et al. examined the use of a probiotic mix, called IRT-5 (*Lactobacillus casei*, *Lactobacillus acidophilus*, *Lactobacillus reuteri*, *B- bifidum*, and *Streptococcus thermophiles*) on autoimmune dry eye [76]. IRT-5 was effective in reducing dry eye symptoms through self-reactive T cell attenuation, suggesting the relevance of this treatment for the Sjogren’s syndrome (SS). The treatment of this disease, related to microbial dysbiosis and the use of IRT-5 probiotics, may benefit from the attenuation of its clinical manifestation of autoimmune dry eye, leading to a clinical improvement of SS. In conclusion, there is an emerging understanding of the effectiveness of probiotics and prebiotics in improving symptoms of eye diseases and in modulating eye inflammation, eye surface health, and homeostasis. Identifying the mechanisms of action of a microbiome-driven systemic approach will allow paving the way for a new generation of rigorous clinical trials to provide alternative and more effective solutions for the management of certain ophthalmic diseases.

## 9. Conclusions

The use of molecular-based techniques, such as next-generation sequencing, allowed the discovery of a vast and diversified commensal bacterial flora that inhabits the human cornea and conjunctiva. The combination of these microorganisms constitutes a real microbiota capable of protecting the ocular surface from colonization by potentially pathogenic microbial species. The most represented bacteria of the ocular microbiota belong to the following genera: *Pseudomonas*, *Propionibacterium*, *Acinetobacter*, and *Corynebacterium*. However, this homeostatic microbiota can be easily altered by environmental factors, pathological states, such as dry eye syndrome, use of antibiotics, infections, such as blepharitis or conjunctivitis, and personal habits, such as excessive and irresponsible use of contact lenses, which may also represent a vehicle for infections by opportunistic pathogenic microorganisms. Therefore, the disruption of the normal eye microbiota can play a significant role as a cofactor in the pathogenesis of ophthalmic diseases. Furthermore, recent studies have also highlighted how the alteration of the microbiota of other body sites can favor the development of ophthalmic pathologies. Indeed, changes in the composition of the oral and intestinal microbiota have been associated with glaucoma, uveitis, and AMD, respectively. The study of the ocular microbiota is important to improve our knowledge of the role of homeostatic microorganisms in the prevention of several ophthalmic diseases and to develop new therapeutic strategies, based above all, on the intake of probiotics, for the treatment of ocular pathologies.

## Figures and Tables

**Figure 1 microorganisms-08-01033-f001:**
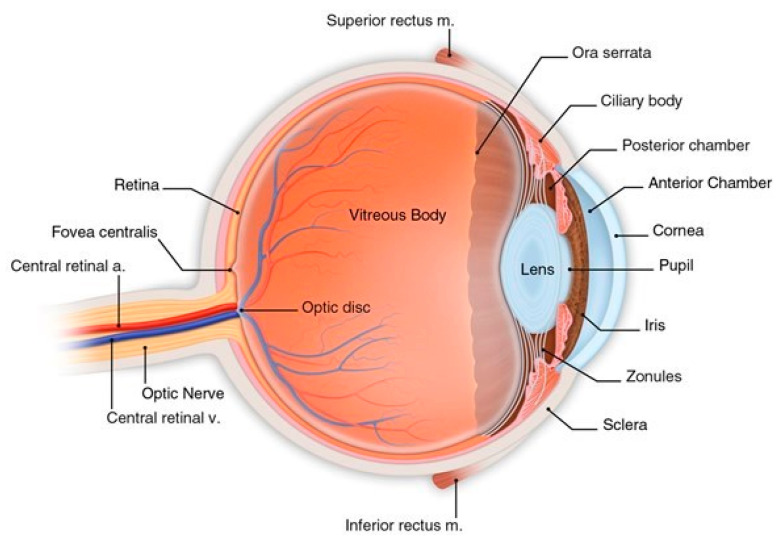
Anatomy of the eye (© 2020 American Academy of Ophthalmology) [12].

**Figure 2 microorganisms-08-01033-f002:**
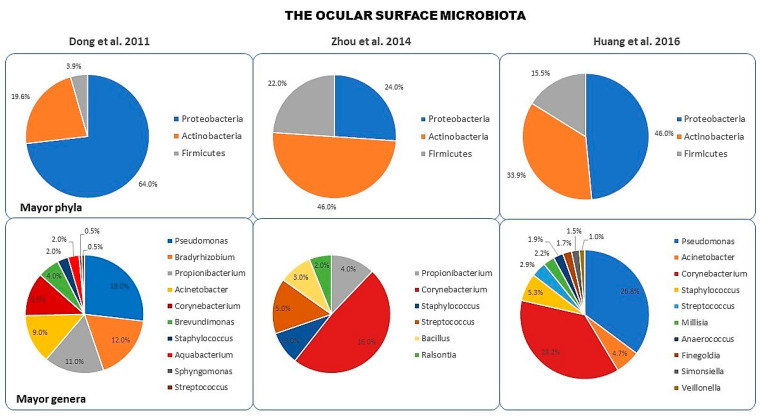
Relative abundance of major phyla and mayor genera found in the ocular surface using 16S rRNA gene reads. Dong et al. analyzed 115,003 sequences in total (2011), Huang et al. analyzed 840,373 high-quality sequencing reads (2016), Zhou et al. generated 1,690,427 reads (2014).

**Table 1 microorganisms-08-01033-t001:** Mayor genera detected in the healthy eye microbiome and changes associated with disease (blepharitis, trachoma, diabetes) and contact lens wearing, detected using 16S rRNA gene sequencing.

Mayor Genera Present in Healthy Eye	Ocular Microbiota Changes Associated with Disease and Contact Lens Wearing
*Corynebacterium*	Increase in blepharitis and trachoma
*Staphylococcus*	Increase in blepharitis and decrease in contact lens wearers
*Streptococcus*	Increase in trachoma and decrease in contact lens wearers
*Propionibacterium*	Decrease in blepharitis
*Pseudomonas*	Increase in contact lens wearers and keratitis
*Acinetobacter*	Increase in contact lens wearers, keratitis, and diabetes

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
