# Peer review of "Current Evidence on the Ocular Surface Microbiota and Related Diseases"

_microorganisms, 2020, doi:10.3390/microorganisms8071033_

Round 1
Reviewer 1 Report
There is a great interest in the role of the ocular surface microbiota and ocular surface diseases; therefore, the review by Petrillo and colleagues is very timely, and it is well written.
There are some concerns about some generalizations without data to support in the literature and some essential concepts that were left out.
Concerns:
1. Lines 44-45. The authors stated, “The ocular microbiota has a role in maintaining the homeostasis of the ocular surface, preventing colonization of pathogenic species.” This is a controversial issue, as even the presence of commensal bacteria on the eye has not been proved. Furthermore, ocular microbiota might influence the ocular surface without preventing the colonization of pathogenic species. In support of this, the group from Baylor has shown that germ-free mice have dry eye (PMID: 27087247; PMID: 29438346).
2. Lines 61-63. The authors stated, “Resident conjunctiva lymphocytes, plasma cells, neutrophils, and dendritic cells can generate inflammatory response and produce antibodies against potential pathogenic environmental microorganisms.” The conjunctiva is not a plasma cell-rich environment, and as it is written, it suggests that everything is happening at the conjunctiva and not in the draining nodes, as it is the current knowledge. The reviewer suggests a revision of the statement.
3. Lines 74-74. The presence of a stable, ocular microbiota is still debatable. Please revise the statement.
4. Line 79. Ref 13 is a review. Can the authors provide a study to support their statement?
5. Line 87. Can the authors revise “In human medicine?”
6. Lines 105-106. The statement is borrowing from the gut microbiome and extrapolating to the ocular surface. It has not been shown that an ocular microbiota may modulate ocular tolerance, as it has been shown in the gut for tolerance to commensal and food. Please provide adequate references for that concept or revise it.
7. Lines 120-125. Ref. 21 is a review. Can the authors cite original studies instead?
8. Lines 125-126. This statement is later contradicted by the authors when they cite the work by Kara Cavuoto. Please revise.
9. Lines 140-145. The first study to investigate bacterial composition using 16S was in 2007 by Grahan and colleagues, and not by Dong. (Ocular Pathogen or Commensal: A PCR-based Study of Surface Bacterial Flora in Normal and Dry Eyes, PMID: 18055811).
10. Line 145. Ref 23 is not by Dong and colleagues.
11. Figure 2 is very blurry. Can the authors provide a replacement with a higher definition image?
12. Lines 190-197. This is a broad statement. While it might be true, there is still a debate if a stable ocular microbiome exits or not. Metagenomics detected only DNA and DNA is long-lived; it does not characterize a stable bacterial community. Furthermore, traditional cultures of conjunctival swabs yield bacterial colonies in less than 50% of the cases and the authors do not even mention this in the review. As it is written, it sounds like a stable microbiome is easy to identify by cultures and metagenomics and neither is true.
13. Part #5. The role of gut intestinal microbiome and ocular surface diseases such as dry eye and Sjogren Syndrome deserves a paragraph. Please also include insights from animal models:PMC7021297; PMID: 32293464; PMID: 27087247; PMID: 29438346; PMID: 29934134
14. Table 1: Please include citations to each of the rows, showing an increase or decrease of the bacteria.
15. Part 7. Please include insights into how antibiotics alter the ocular surface response in animals. Cite: PMID: 31434991; PMID: 30470496; PMID: 31618426
16. Lines 379-382. The authors stated, “Similar to quercetin, Vela et al. examined the effects of resveratrol alone and in combination with quercetin, in mouse models, demonstrating a reduction in clinical signs of DED, with better corneal coloration and anti-inflammatory effects, suggesting their topical application for the treatment of DED [74].” Do the authors suggest that resveratrol or quercetin modulate the ocular microbiome? The reviewer is familiar with that study, and although resveratrol and quercetin might have a beneficial effect on dry eye disease, the citation of that work on this review is not appropriate.
Author Response
Cover Letter in Response to Reviewer’s Comments
Name of journal: Microorganisms (ISSN 2076-2607)
Manuscript ID: microorganisms-852370
Title: Current evidences on ocular surface microbiota and related diseases
Reviewer 1
- Lines 44-45. The authors stated, “The ocular microbiota has a role in maintaining the homeostasis of the ocular surface, preventing colonization of pathogenic species.” This is a controversial issue, as even the presence of commensal bacteria on the eye has not been proved. Furthermore, ocular microbiota might influence the ocular surface without preventing the colonization of pathogenic species. In support of this, the group from Baylor has shown that germ-free mice have dry eye (PMID: 27087247; PMID: 29438346).
- Lines 61-63. The authors stated, “Resident conjunctiva lymphocytes, plasma cells, neutrophils, and dendritic cells can generate inflammatory response and produce antibodies against potential pathogenic environmental microorganisms.” The conjunctiva is not a plasma cell-rich environment, and as it is written, it suggests that everything is happening at the conjunctiva and not in the draining nodes, as it is the current knowledge. The reviewer suggests a revision of the statement.
- Lines 74-74. The presence of a stable, ocular microbiota is still debatable. Please revise the statement.
- Line 79. Ref 13 is a review. Can the authors provide a study to support their statement?
- Line 87. Can the authors revise “In human medicine?”
- Lines 105-106. The statement is borrowing from the gut microbiome and extrapolating to the ocular surface. It has not been shown that an ocular microbiota may modulate ocular tolerance, as it has been shown in the gut for tolerance to commensal and food. Please provide adequate references for that concept or revise it.
- Lines 120-125. Ref. 21 is a review. Can the authors cite original studies instead?
- Lines 125-126. This statement is later contradicted by the authors when they cite the work by Kara Cavuoto. Please revise.
- Lines 140-145. The first study to investigate bacterial composition using 16S was in 2007 by Grahan and colleagues, and not by Dong. (Ocular Pathogen or Commensal: A PCR-based Study of Surface Bacterial Flora in Normal and Dry Eyes, PMID: 18055811).
- Line 145. Ref 23 is not by Dong and colleagues.
- Figure 2 is very blurry. Can the authors provide a replacement with a higher definition image?
- Lines 190-197. This is a broad statement. While it might be true, there is still a debate if a stable ocular microbiome exits or not. Metagenomics detected only DNA and DNA is long-lived; it does not characterize a stable bacterial community. Furthermore, traditional cultures of conjunctival swabs yield bacterial colonies in less than 50% of the cases and the authors do not even mention this in the review. As it is written, it sounds like a stable microbiome is easy to identify by cultures and metagenomics and neither is true.
- Part #5. The role of gut intestinal microbiome and ocular surface diseases such as dry eye and Sjogren Syndrome deserves a paragraph. Please also include insights from animal models:PMC7021297; PMID: 32293464; PMID: 27087247; PMID: 29438346; PMID: 29934134
- Table 1: Please include citations to each of the rows, showing an increase or decrease of the bacteria.
- Part 7. Please include insights into how antibiotics alter the ocular surface response in animals. Cite: PMID: 31434991; PMID: 30470496; PMID: 31618426
- Lines 379-382. The authors stated, “Similar to quercetin, Vela et al. examined the effects of resveratrol alone and in combination with quercetin, in mouse models, demonstrating a reduction in clinical signs of DED, with better corneal coloration and anti-inflammatory effects, suggesting their topical application for the treatment of DED [74].” Do the authors suggest that resveratrol or quercetin modulate the ocular microbiome? The reviewer is familiar with that study, and although resveratrol and quercetin might have a beneficial effect on dry eye disease, the citation of that work on this review is not appropriate.
Author Response to reviewer 1
- A. Lines 44-45. The authors stated, “The ocular microbiota has a role in maintaining the homeostasis of the ocular surface, preventing colonization of pathogenic species.” This is a controversial issue, as even the presence of commensal bacteria on the eye has not been proved. Furthermore, ocular microbiota might influence the ocular surface without preventing the colonization of pathogenic species. In support of this, the group from Baylor has shown that germ-free mice have dry eye (PMID: 27087247; PMID: 29438346).
- R. line 44-45 has been replaced with: The ocular microbiota has a role in maintaining and preservation of the health of the ocular surface.
- A. Lines 61-63. The authors stated, “Resident conjunctiva lymphocytes, plasma cells, neutrophils, and dendritic cells can generate inflammatory response and produce antibodies against potential pathogenic environmental microorganisms.” The conjunctiva is not a plasma cell-rich environment, and as it is written, it suggests that everything is happening at the conjunctiva and not in the draining nodes, as it is the current knowledge. The reviewer suggests a revision of the statement.
- R. Lines 61-63 has been replaced with: Resident conjunctiva lymphocytes can generate inflammatory response and produce antibodies against potential pathogenic environmental microorganisms.
- A. Lines 74-74. The presence of a stable, ocular microbiota is still debatable. Please revise the statement.
- R. Lines 74-75 have been cancelled.
- A. Line 79. Ref 13 is a review. Can the authors provide a study to support their statement?
- R. Ref 13 has been replaced with reference: Modulation of Corneal Epithelial Innate Immune Response to Pseudomonas Infection by Flagellin Pretreatment (PMID 17898290)
- A. Line 87. Can the authors revise “In human medicine?”
- R. Line 87. This sentence was modified as follows: In human, 13 different TLRs exist and each has its own specificity.
- A. Lines 105-106. The statement is borrowing from the gut microbiome and extrapolating to the ocular surface. It has not been shown that an ocular microbiota may modulate ocular tolerance, as it has been shown in the gut for tolerance to commensal and food. Please provide adequate references for that concept or revise it.
- R. Lines 105 – 106. This sentence - Indeed, the activation is appropriate if it is directed against disabling pathogens, but it can become inappropriate if directed to the normal microbial flora. - have been cancelled.
- A. Lines 120-125. Ref. 21 is a review. Can the authors cite original studies instead?
- R. Lines 120-125. Ref 21 has been replaced with reference: Paucibacterial Microbiome and Resident DNA Virome of the Healthy Conjunctiva (27699405)
- A. Lines 125-126. This statement is later contradicted by the authors when they cite the work by Kara Cavuoto. Please revise.
- R. Lines 125-126. This sentence - This ecosystem remains relatively stable throughout life unless it is altered by antibiotic treatment, surgical interventions, infections, or other problems (e.g. use of contact lenses in human medicine) - have been cancelled.
- A. Lines 140-145. The first study to investigate bacterial composition using 16S was in 2007 by Grahan and colleagues, and not by Dong. (Ocular Pathogen or Commensal: A PCR-based Study of Surface Bacterial Flora in Normal and Dry Eyes, PMID: 18055811).
- R. Lines 140-145. This information was included in the manuscript: The first study to investigate bacterial composition using the sequencing of the 16S rRNA gene was in 2007 by Grahan et al., in which atypical bacteria were identified on the ocular surface including Rhodococcus erythropol, Klebsiella oxytoca and Erwinia sp (18055811). In 2011, Dong et al. have classified the bacteria of four subject’s ocular surface in 5 phyla and 59 genera.
- A. Line 145. Ref 23 is not by Dong and colleagues.
- R. Line 145. The reference has been replaced with the correct one: (21571682)
- A. Figure 2 is very blurry. Can the authors provide a replacement with a higher definition image?
- R. Figure 2 has been improved, increasing the sharpness and image quality
- A. Lines 190-197. This is a broad statement. While it might be true, there is still a debate if a stable ocular microbiome exits or not. Metagenomics detected only DNA and DNA is long-lived; it does not characterize a stable bacterial community. Furthermore, traditional cultures of conjunctival swabs yield bacterial colonies in less than 50% of the cases and the authors do not even mention this in the review. As it is written, it sounds like a stable microbiome is easy to identify by cultures and metagenomics and neither is true.
- R. Lines 190-157. This sentence was modified as follows: The opportunistic and environmental microorganisms, are represent transient species of the ocular surface such as Rothia, Herbaspirillum, Leptothrichia, and Rhizobium. On the contrary, Staphylococci, Cornyebacteriae, and Proteobacteria represent the main species no transient of the ocular surface.
- A. Part #5. The role of gut intestinal microbiome and ocular surface diseases such as dry eye and Sjogren Syndrome deserves a paragraph. Please also include insights from animal models:PMC7021297; PMID: 32293464; PMID: 27087247; PMID: 29438346; PMID: 29934134
- R. We are thanks to the reviewer for the suggestion. The focus of our next review will be the connection between intestinal and ocular microbiota connected to ocular surface diseases. We reserve this suggestion for the next study
- A. Table 1: Please include citations to each of the rows, showing an increase or decrease of the bacteria.
- R. We agree with the reviewer. The citations have been included.
- A. Part 7. Please include insights into how antibiotics alter the ocular surface response in animals. Cite: PMID: 31434991; PMID: 30470496; PMID: 31618426
- R. We have viewed the recommended articles and show correlation between intestinal microbiota and eye diseases. In the paragraph reported, we treated changes in the ocular microbiota only in response to specific ocular treatments (eye drops). Similarly, we reserve this suggestion for the next study.
- A. Lines 379-382. The authors stated, “Similar to quercetin, Vela et al. examined the effects of resveratrol alone and in combination with quercetin, in mouse models, demonstrating a reduction in clinical signs of DED, with better corneal coloration and anti-inflammatory effects, suggesting their topical application for the treatment of DED [74].” Do the authors suggest that resveratrol or quercetin modulate the ocular microbiome? The reviewer is familiar with that study, and although resveratrol and quercetin might have a beneficial effect on dry eye disease, the citation of that work on this review is not appropriate.
- R. Lines 379-382 have been cancelled.
Reviewer 2 Report
The paper is an interesting piece on ocular surface microbiota and I think it is worth publishing. In my opinion, some additions are needed in the section 2 (Eye structure and immunity). Authors should mention, that almost every element of the eye is different in the immune response, due to the presence of different immunological cells and proteins (mucins, defensins, cathelicidin) and impact of immune cells. After this addition the paper may be published.
Author Response
Cover Letter in Response to Reviewer’s Comments
Name of journal: Microorganisms (ISSN 2076-2607)
Manuscript ID: microorganisms-852370
Title: Current evidences on ocular surface microbiota and related diseases
Reviewer 2
- The paper is an interesting piece on ocular surface microbiota and I think it is worth publishing. In my opinion, some additions are needed in the section 2 (Eye structure and immunity). Authors should mention, that almost every element of the eye is different in the immune response, due to the presence of different immunological cells and proteins (mucins, defensins, cathelicidin) and impact of immune cells. After this addition the paper may be published.
Author Response to reviewer 2
- A. The paper is an interesting piece on ocular surface microbiota and I think it is worth publishing. In my opinion, some additions are needed in the section 2 (Eye structure and immunity). Authors should mention, that almost every element of the eye is different in the immune response, due to the presence of different immunological cells and proteins (mucins, defensins, cathelicidin) and impact of immune cells. After this addition the paper may be published.
- R. Dear reviewer, thank you for the positive opinion and for the suggestions received. We proceeded to add further details within the second paragraph as suggested (lines 59-71). I hope they can satisfy what is requested.
Round 2
Reviewer 1 Report
The authors have been responsive; no further comments.